# Verubulin (Azixa) Analogues with Increased Saturation: Synthesis, SAR and Encapsulation in Biocompatible Nanocontainers Based on Ca^2+^ or Mg^2+^ Cross-Linked Alginate

**DOI:** 10.3390/ph16101499

**Published:** 2023-10-21

**Authors:** Kseniya N. Sedenkova, Denis N. Leschukov, Yuri K. Grishin, Nikolay A. Zefirov, Yulia A. Gracheva, Dmitry A. Skvortsov, Yanislav S. Hrytseniuk, Lilja A. Vasilyeva, Elena A. Spirkova, Pavel N. Shevtsov, Elena F. Shevtsova, Alina R. Lukmanova, Vasily V. Spiridonov, Alina A. Markova, Minh T. Nguyen, Alexander A. Shtil, Olga N. Zefirova, Alexander A. Yaroslavov, Elena R. Milaeva, Elena B. Averina

**Affiliations:** 1Department of Chemistry, Lomonosov Moscow State University, 119991 Moscow, Russia; sedenkova@med.chem.msu.ru (K.N.S.); denis.leshchukov@chemistry.msu.ru (D.N.L.); grishin@nmr.chem.msu.ru (Y.K.G.); kolaz92@gmail.com (N.A.Z.); jullina74@mail.ru (Y.A.G.); skvorratd@mail.ru (D.A.S.); gritseniuk2000@yandex.ru (Y.S.H.); lukmanovaalina@mail.ru (A.R.L.); vasya_spiridonov@mail.ru (V.V.S.); olgaz_13@mail.ru (O.N.Z.); yaroslav@belozersky.msu.ru (A.A.Y.); milaeva@med.chem.msu.ru (E.R.M.); shtilaa@yahoo.com (A.A.S.); 2Faculty of Bioengineering and Bioinformatics, Lomonosov Moscow State University, 119991 Moscow, Russia; liljavasilyeva@gmail.com; 3Institute of Physiologically Active Compounds at Federal Research Center of Problems of Chemical Physics and Medicinal Chemistry, Russian Academy of Sciences (IPAC RAS), 142432 Chernogolovka, Russia; kustova.ea@mail.ru (E.A.S.); pnshevtsov@gmail.com (P.N.S.); shevtsova@ipac.ac.ru (E.F.S.); 4Emanuel Institute of Biochemical Physics, Russian Academy of Sciences, 119334 Moscow, Russia; alenmark25@gmail.com (A.A.M.); tuantonyx@yahoo.com (M.T.N.); 5Institute of Cyber Intelligence Systems, National Research Nuclear University MEPhI, 115409 Moscow, Russia

**Keywords:** tubulin polymerization inhibitor, apoptosis inducer, verubulin, pyrimidine, tetrahydroquinazoline, aromatic nucleophilic substitution, nanocontainer, cross-linked alginate

## Abstract

Tubulin-targeting agents attract undiminished attention as promising compounds for the design of anti-cancer drugs. Verubulin is a potent tubulin polymerization inhibitor, binding to colchicine-binding sites. In the present work, a series of verubulin analogues containing a cyclohexane or cycloheptane ring 1,2-annulated with pyrimidine moiety and various substituents in positions 2 and 4 of pyrimidine were obtained and their cytotoxicity towards cancer and non-cancerous cell lines was estimated. The investigated compounds revealed activity against various cancer cell lines with IC_50_ down to 1–4 nM. According to fluorescent microscopy data, compounds that showed cytotoxicity in the MTT test disrupt the normal cytoskeleton of the cell in a pattern similar to that for combretastatin A-4. The hit compound (*N*-(4-methoxyphenyl)-*N*,2-dimethyl-5,6,7,8-tetrahydroquinazolin-4-amine) was encapsulated in biocompatible nanocontainers based on Ca^2+^ or Mg^2+^ cross-linked alginate and it was demonstrated that its cytotoxic activity was preserved after encapsulation.

## 1. Introduction

Tubulin is a globular protein essential for the functioning of all eukaryotic cells. Heterodimers composed of α- and β-tubulin molecules polymerize to form microtubules, which play a key role in the formation of cytoskeleton, cell division and intracellular transport [1,2,3]. Tubulin- and microtubule-targeting agents have been the subject of intensive research and some of them are now widely used both in chemotherapy and as research tools to study the details of microtubule functioning. Despite widespread clinical use of anti-microtubule drugs, research in the field is constantly ongoing and associated mainly with the search for new alternatives with lower toxicities, better bioavailability and effectivity against resistant tumors [4,5,6,7,8,9,10].

The binding of tubulin-targeting agents to different regions of α,β-tubulin dimer causes its conformation switch and regulates microtubule behavior [3,4]. The compounds interacting with the taxol-binding site and stabilizing microtubules [11], as well as microtubule-destabilizing agents that bind to the vinca domain of tubulin [12], are used in chemotherapy. Several representatives of a large group of colchicine-binding site ligands have also been introduced into clinical trials, the most successful of which was fosbretabulin (combretastatin A-4 prodrug) [13]. However, so far none of these compounds has received full and final approval for clinical use. Considering the fact that many ligands of the colchicine-binding site are both efficient and easily accessible synthetically, they attract significant interest of researchers, and extensive efforts are being made to develop new molecules with this type of action [13,14,15]. Note that the common limitations for the use of these compounds as drug candidates are their low selectivity towards non-tumor cells (resulting in high general toxicity) and, for certain molecules, poor bioavailability.

One of the colchicine site ligands that has successfully passed a phase II clinical trial is verubulin (Azixa, **1**) [16,17,18,19] (Figure 1). Since the first reports of verubulin as a potent microtubule-disrupting agent and especially after the suspension of its further clinical promotion due to economic reasons [20], several research groups extensively studied the structure–activity relationship for the series of analogues of the lead molecule (Figure 1) [20,21,22,23,24,25,26,27,28,29,30,31,32,33,34,35,36]. In particular, the following modifications were carried out: the variations of the substituent in position *2* of quinazoline, conformational restriction of 4-methoxy-*N*-methylaniline moiety, replacement of quinazoline core with other aromatic heterocycles, etc.

In 2021, Banerjee et al. [33] showed that replacement of the fused benzene ring in quinazoline for a saturated 5-membered ring can lead to compounds with high cytotoxicity (see example in Figure 1). Recently, J. Tan et al. [34] described a large series of verubulin analogues, in one of which the quinazoline core was replaced with tetrahydroquinazoline (Figure 1). This compound not only retained high activity but also demonstrated the best selectivity (by 4–5 times) for tumor cells, despite the fact that most of the active substances in the series were not selective at all [34]. The result obtained seemed interesting in the context of the strategy of designing pharmaceutically relevant molecules, including less toxic ones, by increasing their three-dimensionality (“escaping the flatland”), which has been widely discussed in recent decades [37,38,39,40]. Therefore, it seemed interesting to conduct a systematic study of verubulin analogues with a reduced degree of flatness due to the replacement of the benzene ring fused with pyrimidine for a six- or seven-membered saturated ring **B** (Figure 1). In the present work, we synthesized a series of such analogues, also varying their conformational rigidity and the substituents in positions 2 and 4 of heterocyclic core **A** (Figure 1), and studied them in cell viability assays on cell lines of cancer and non-cancerous etiology and other biotests. 

Additionally, we explored the possibility to improve the toxicological profile of a selected compound via its encapsulation into nanoparticles formed from Ca^2+^ or Mg^2+^ cross-linked alginate biopolymers. Alginate is a biodegradable and biocompatible copolymer of guluronic and mannuronic acids, which has already received US Food and Drug Administration (FDA) approval for medical applications [41]. This natural anionic polymer (biomaterial) has been widely investigated and used in many areas of biomedical science and technology due to its biocompatibility, low toxicity, relatively low cost, softness and easy gelation upon addition of divalent cations (Ca^2+^ and Mg^2+^) [42]. The role of Ca^2+^ and Mg^2+^ ions is to form a framework structure of polysaccharide macromolecules, accompanied by a significant compactization of the initial macromolecules of alginate. Alginate has promising biopharmaceutical properties such as pH sensitivity, biocompatibility, biodegradability, mucoadhesiveness, lack of toxicity and immunogenicity, which make it attractive for the development of carriers for drug immobilization [43]. In our previous work, we have demonstrated that encapsulation of an isoxasole derivative with anti-cancer activity in alginate biopolymer can significantly increase its selectivity towards cancer cells [44].

## 2. Results and Discussion

### 2.1. Chemistry

4-Aminopyrimidines **2a**–**r** were obtained via nucleophilic substitution of chlorine in the corresponding 4-cloropyrimidines upon the treatment with secondary amines in the presence of a catalytic amount of HCl (Figure 1 and Figure 2). In most cases, the reactions of 4-chloropyrimidines **3a**–**c** with amines proceeded smoothly, affording target heterocycles in moderate to good yields.

The reaction of 2,4-dichlorotetrahydroquinazoline (**3d**) with 4-methoxy-*N*-methylaniline led to a multicomponent mixture of product, from which only the prevailing product **2n** resulted from two-fold substitution and compound **2m** could be isolated in low yields (Figure 2). By reducing the time and using the excess of dichloride **3d**, we managed to obtain tetrahydroquinazoline **2m** as the main product, though during the purification the yield of this compound decreased. The interaction of dichloride **3d** and sterically hindered 6-methoxytetrahydroquinoline was even more complicated and required a long reflux, after which the compounds **2o**–**r** could be isolated from the reaction mixture (Figure 2). The structure of all the obtained compounds was unambiguously determined via NMR spectroscopy, using 2D techniques, when necessary (see Appendix A). 

### 2.2. Bioactivity Testing

#### 2.2.1. Cytotoxicity of Verubulin Analogues and SAR Analysis

The obtained series of verubulin analogues **2a**–**r** (which also included three by-products isolated in the reactions according to Figure 2) were initially evaluated for cytotoxicity against commonly used human breast cancer cell line MCF7′ (fast-growth subclone), human lung epithelial carcinoma cell line A549 and etiologically non-cancer immortalized lung fibroblast cell line VA13 (WI38 subline 2RA). Cytotoxicity against immortalized human embryonic kidney cell line HEK293T, characterized by a high growth rate, was also studied regarding cytotoxicity against non-cancerous fast-growth cells. A standard MTT assay [45] was performed; verubulin (**1**) served as a positive reference. The data obtained are presented in Table 1 (for dose–response curves, see Appendix A).

Saturated verubulin derivative **2c** retained high cytotoxicity of the parent molecule (**1**) similarly to its analogue with a chlorine substituent in position 2 of ring **A** (**2m**), described by Tan et al. [34]. Compounds **2c** and **2m** are more cytotoxic than their counterpart with a more polar amino group at C2 (**2i**), which is consistent with molecular dynamics modeling data (see below). Increasing the size of the alicycle **B** in the verubulin analogue **2c** to a seven-membered ring (**2k**) reduces cytotoxicity by more than 10 times. A noticeable drop in activity during the expansion of ring **B** is most likely due to the steric hindrances it creates. Thus, this result denotes a limitation imposed on the volume of alicyclic ring **B**.

Saturated verubulin analogues (**2c**, **2k**), regardless of the size of ring **B**, demonstrated no selectivity towards tumor cells in the pair A549/VA13. It should be noted that verubulin and its saturated analogue **2m** which has been reported to be selective in the pair MCF7/MCF10A [34] demonstrated no selectivity in the pair A549/VA13 either. 

Conformational restriction of 4-methoxy-*N*-methylaniline moiety in saturated verubulin derivative **2c** led to compound **2e** with slightly less activity, but with an IC_50_ still in the nanomolar concentration range. As seen in Figure 1, a similar change in activity was observed for the rigid analogue of verubulin obtained by Wang et al. [25]. The same modification for verubulin analogue **2k** with a seven-membered ring **B** resulted in compound **2l**, which was seven to twelve times less active. Thus, the cytotoxicity in the series of both “open” and “rigid” compounds, depending on ring **B**, changed as follows: benzene > cyclohexane > cycloheptane.

In a series of conformationally restricted tetrahydroquinazolines, the important role of a non-polar substituent at position 2 of ring **A** was also demonstrated: the replacement of the methyl group by an amino group slightly reduced cytotoxicity against cancer cells (**2j** vs. **2e**). Similar replacement with a hydroxyl group led to a complete loss of cytotoxicity (**2o** vs. **2e**), which was, however, largely restored if the hydroxyl was converted to non-polar ether (O-*i*Pr) moiety (**2q** vs. **2o**).

Finally, analysis of SAR in the series of obtained verubulin analogues shows the critical role of the para-methoxy substituent in ring **C**. Its removal reduced cytotoxicity by 2–3 orders of magnitude (**2a** vs. **2c**, **2g** vs. **2i**), while its replacement with the methyl group was less detrimental, although it still reduced the activity by an order of magnitude (**2b** vs. **2c**, **2j** vs. **2h**). Compound **2f** containing a meta-nitro group in ring **C** was inactive. Miscellaneous compounds **2n**, **2p** and **2r** obtained as by-products in the synthetic schemes were moderately cytotoxic.

#### 2.2.2. Effect of Verubulin Analogues on Microtubule Dynamics

Since verubulin is known to stimulate microtubule disassociation, we performed an immunofluorescent microscopy study to determine the response of microtubule dynamics in carcinoma A549 cells to the treatment with representative compounds **2b**,**c**,**e**. As shown in Figure 2, exposure to 5–10 μM of the compounds for 24 h leads to a pronounced or complete disruption of microtubules (Figure 2A–C), and the pattern of a diffuse distribution of the stained microtubule subunits corresponds to that caused by combretastatin CA-4 (Figure 2E). Aberrant cell morphology—contraction and rounding (Figure 2A,B) or contraction and elongation (Figure 2C)—was also observed. As expected, immunofluorescent staining of microtubules in the cells treated with compound **2a**, characterized by low cytotoxicity, revealed no effect on the microtubule cytoskeleton even at very high concentrations of this compound (Figure 2D).

An additional study of the effect of compounds **2c** and **2e** on microtubule assembly in vitro was monitored by recording the change in optical density at 355 nm during the GTP-dependent polymerization on crude preparation of tubulin (Tb) and microtubule-associated proteins (MAPs) from mouse brains. The obtained graphs of the change in optical density versus time are shown in Figure 3. It was shown that both **2c** and **2e** inhibited the initial rate of tubulin polymerization in a dose-dependent manner similarly to verubulin. The concentrations of the compounds were the same as in Figure 2. On the basis of initial rates (from 6 to 16 min for control and from 12 to 32 min for others) of the linear change in optical density at 355 nm vs. compound concentrations, where each point is the average of two different experiments with three repeats in each, the IC_50_ were calculated for **2c**, **2e** and verubulin: 7.2 ± 1.9 μM; 26 ± 6 μM and 3.1 ± 1.1 μM, respectively (Appendix A). It should be mentioned that unlike control probe (DMSO) in the presence of **2c**, **2e** and verubulin there are no features of depolymerization which causes a plateau in the control curve (Figure 3). 

#### 2.2.3. Molecular Dynamics Simulations for Compound **2c**

The obtained data (Figure 2 and Figure 3) confirm that new compounds significantly disturb the assembly of microtubules and exhibit characteristics similar to tubulin polymerization inhibitors, particularly such as verubulin. As was mentioned above, the latter binds to the colchicine site of tubulin, which is located in the β-subunit at the interface with the α-subunit of the α,β-heterodimer. The data of X-ray diffraction analysis of tubulin complexes with verubulin (PDB ID: 5XKF) [46] and several of its analogues [29,30,33,34] have been collected recently. To get an idea of how the saturated analogue of verubulin **2c** (the most active in the series) binds to the protein, we performed molecular dynamics simulations (Figure 4). The plot of the root mean square deviations (RMSDs, Figure 4A) for the protein, GDP, GTP and ligand non-hydrogen atoms and the visual analysis of the trajectory confirm that system stability is retained from about the fiftieth nanosecond until the end of the course of the production simulation (100 ns), although the position of the ligand is slightly adjusted compared to the docking pose (by a shift of tetrahydroquinazoline residue).

As shown in Figure 4B, the saturated analogue of verubulin, compound **2c**, is oriented in the colchicine-binding site similar to the parent molecule. Ring **B** in **2c** is located in the hydrophobic pocket within the side chains of Ala252β, Ala314β, Ala315β, Ile316β. Both the methyl group attached to tetrahydroquinazoline and the methoxy group of 4-methoxyphenyl residue also form important hydrophobic interactions with the β-tubulin subunit (the former is exposed in the hydrophobic subpocket formed by the side chain of Leu250β, Leu253β and the aliphatic chain of Gln245, while the latter is located close to the aliphatic chains of Met257β and Lys350β). The N^1^ atom of ring **A** of verubulin analogues, as was shown by Banerjee et al. [29], can form a key water-mediated hydrogen bond with the main chain of Cys239β, and since such an interaction is water-mediated it is theoretically possible for compound **2c**, despite some differences in the location of ring **A** compared to verubulin.

The binding mode of compound **2c** corresponds to the observed SAR patterns, in particular, to a decrease in activity when the non-polar group at position 2 of ring **A** was replaced by a polar one, as well as a loss of activity upon removal of the methoxy group in ring **C** (see above, Table 1). The dramatic difference in cytotoxicity of compounds **2o** and **2j** (both with polar substituents at position 2 of ring **A**) is most likely due to the disruption of the above-mentioned water-mediated hydrogen bonds of the N*^1^* with the key amino acid residue Cys239β caused by the hydroxyl group of **2o**, capable of forming stronger hydrogen bonds than the amino group of its counterpart **2j**. Overall, molecular modeling and SAR data further confirm that saturated verubulin analogue **2c** retains the mode of action of the parent molecule.

#### 2.2.4. Apoptosis Induction by Verubulin Analogues

Apoptosis induction by verubulin and hit compounds **2c**,**e**,**i**,**j**,**m** was studied by flow cytometry to reveal the mechanism of HCT116 tumor cell death. Apoptosis is an important and active regulatory pathway of cell growth and proliferation. *Annexin V* is a calcium-dependent phospholipid-binding protein with a high affinity for phosphatidylserine (PS), a membrane component normally localized to the internal face of the cell membrane. Early in the apoptotic pathway, molecules of PS are translocated to the outer surface of the cell membrane where *Annexin V* can readily bind them [47,48].

All tested compounds were found to significantly induce apoptosis of HCT116 cells after 48 h of exposure (Figure 5). Nearly 77.5% of untreated HCT116 cells were viable (*Annexin V* (−) and 7-AAD (−)). For verubulin (**1**), the fraction of viable cells was reduced compared to the control, and almost half of the studied cells (49.3%) were found at the early (*Annexin V* (+) and 7-AAD (−) 16.3%) or late (*Annexin V* (+) and 7-AAD (+) 33%) apoptosis stages. The number of viable cells was about 50% of the entire studied population of HCT116 cells. Saturation of ring B (**2c**) was shown to affect the activation of apoptosis: the population of apoptotic cells decreased down to 30% when treated with **2c** (Figure 5D). Further conformational restriction (**2e**) led to a decrease in the number of viable cells down to 30% and a significant increase in the number of cells at the late apoptosis stage (Figure 5E). The replacement of the methyl group in position 2 of the pyrimidine ring in compound **2c** by chlorine (**2m**) had no effect on the apoptotic pattern compared to verubulin, while its replacement by a NH_2_ group (**2i**) changed it: the percentage of early apoptotic cells increased significantly (46.3%) and the population of cells in late apoptosis decreased. Thus, the introduction of an amino group into the pyrimidine ring affects the rate of cell entry into apoptosis. For compound **2j**, the population of cells in early apoptosis increased up to 56.4%.

The obtained data demonstrate that saturation of ring **B** in the verubulin molecule, though preserving both cytotoxicity and the effect on microtubules, does not lead to an increase in the selectivity. Therefore, we studied the opportunity to improve the toxicological profile for compound **2c** via its encapsulation in Ca^2+^/Mg^2+^ cross-linked alginate nanoparticles as a drug delivery system.

### 2.3. Encapsulation of Compound **2c** in Biocompatible Nanocontainers Based on Ca^2+^/Mg^2+^ Cross-Linked Alginate and Investigation of Cytotoxicity of Formulations via MTT Assay

Compound **2c** was encapsulated in alginate (Alg)-based nanocontainers cross-linked with Ca^2+^ and Mg^2+^ ions synthesized following the procedure described in [44,49]. The molar ratio both for [Alginate monomer units]/[Ca^2+^] and [Alginate monomer units]/[Mg^2+^] was 10 ÷ 1. After purification and lyophilization, white complexes were obtained which were dissolved in water that resulted in (Alg)–Ca^2+^ and (Alg)–Mg^2+^ nanocontainer solutions, respectively.

Aminopyrimidine **2c** was transformed into a protonated form **2c**∙HCl for better interaction with the carboxyl groups of anionic nanocontainers (see Appendix A). 

The formation of polysaccharide nanocontainers as well as their interaction with aminopyrimidine **2c**∙HCl was controlled by means of dynamic light scattering (DLS), laser microelectrophoresis and UV spectrophotometry (Appendix A). 

The synthesized nanocontainers as well as nanocontainers filled with **2c** were easily dispersed in bi-distilled water. In all cases, dynamic light scattering found one type of composite microgel with narrow particle size distribution (see Appendix A for typical distribution functions). It was established that the effective hydrodynamic diameter (D*_h_*) of the initial alginate macromolecules in an aqueous solution is 680 nm. The interaction of linear alginate both with Ca^2+^ and Mg^2+^ ions leads to a significant contraction in particle size: for samples (Alg)–Ca^2+^ and (Alg)–Mg^2+^, the particle diameters are 150 nm and 205 nm, respectively (Table 2). The effective diameters of **2c**∙HCl–(Alg)–Ca^2+^ and **2c**∙HCl–(Alg)–Mg^2+^ decrease down to 130 nm and 106 nm, respectively, that confirms the formation of the formulations. The polydispersity of alginate-based nanocarriers was evaluated using PDI [50]. It was shown that the synthesized nanocontainers are characterized by PDI values not exceeding 0.05. Negative EPM values indicate that both **2c**∙HCl–(Alg)–Ca^2+^ and **2c**∙HCl–(Alg)–Mg^2+^ are characterized by high aggregative stability.

The incorporation of aminopyrimidine **2c** into microgels (entries 1 and 3) was also confirmed using UV spectroscopy (Appendix A). The absorption spectrum of protonated aminopyrimidine **2c** shows two pronounced electronic transitions at wavelengths of 225 nm and 300 nm. In the absorption spectra of the interaction products of both microgels with **2c**, there is a sharp decrease in the peak intensity at 225 nm. The peak intensity at 300 nm does not change significantly. The obtained result indicates the electrostatic interaction of the protonated form of **2c** with the carboxyl groups of the microgels [51,52]. In turn, the preservation of the peak at 300 nm can be used for the quantitative determination of the **2c** content in microgels. A calibration curve (Appendix A) was plotted to quantify the **2c** content in the nanocontainers. It was established that both (Alg)–Ca^2+^ and (Alg)–Mg^2+^ bound 16 wt.% aminopyrimidine **2c** that corresponds to the quantitative binding of **2c**. In addition, the nature of the cross-linking ions does not affect the binding capacity of nanocontainers towards aminopyrimidine **2c**.

To evaluate the ability of the nanocontainers to retain the loaded aminopyrimidine **2c**, external dialysis solutions after purification of the microgels were analyzed by measuring the optical density at two wavelengths of 225 nm (D_225_) and 300 nm (D_300_), corresponding to the two maxima in the spectrum of the control aminopyrimidine solution (Appendix A). The D_225_ and D_300_ in external dialysis solutions after purification of systems **2c**∙HCl–(Alg)–Ca^2+^ and **2c**∙HCl–(Alg)–Mg^2+^ were found to be 0.18 and 0.012, respectively, which is comparable to the corresponding value for the solvent (bi-distilled water). That confirms that the loaded compound is retained in the nanocontainers.

The MTT assay was carried out on the A549 cell line and non-tumor hTERT-immortalized human fibroblasts for 48 h (Table 2; for cell survival curves, see Appendix A). While (Alg)–Ca^2+^ and (Alg)–Mg^2+^ nanocontainers demonstrated no or low toxicity against cell lines of different etiology (entries 1 and 3, respectively), their formulations with compound **2c** were found to preserve cytotoxic activity at the level of free amine **2c**.

The differences in cell survival were demonstrated on non-tumor human hTERT-immortalized fibroblasts. While for the Mg^2+^ formulation (entry 3) IC_50_ values obtained for A549 and hTERT-immortalized fibroblast cell lines were the same, the Ca^2+^ formulation (entry 1) demonstrated two-fold selectivity towards tumor cells.

The data obtained allow us to conclude that the cytotoxic activity against the A549 cell line is preserved under the action of the compound **2c** in alginate formulations compared to its free form. The encapsulation of the amine in (Alg)–Ca^2+^ nanoparticles is more promising from the point of view of selectivity to cancer cells.

## 3. Materials and Methods

### 3.1. Chemistry

#### 3.1.1. General Remarks

^1^H and ^13^C NMR spectra were recorded on a 400 MHz Agilent 400-MR spectrometer (400.0 and 100.6 MHz for ^1^H and ^13^C respectively; Agilent Technologies, Santa Clara, CA, USA); chemical shift δ were measured with reference to the solvent (CDCl_3_, δ_H_ = 7.26 ppm, δ_C_ = 77.16 ppm). When necessary, assignments of signals in NMR spectra were made using 2D techniques. Fourier-transform infrared (FT-IR) spectra were obtained on a Bruker ALPHA FT-IR spectrometer (Bruker Daltonik GmbH, Bremen, Germany). Accurate mass measurements (HRMS) were performed on a Bruker micrOTOF II mass spectrometer (Bruker Daltonik GmbH, Bremen, Germany) with electrospray ionization (ESI). Mass spectra (MS) were obtained using a GC-MS Thermo ISQ instrument (Thermo Fisher Scientific, Waltham, MA, USA) with electron ionization (EI). Analytical thin layer chromatography was carried out with silica gel plates supported on aluminum (ALUGRAM^®^ Xtra SIL G/UV_254_, Macherey-Nagel, Duren, Germany); the detection was carried out by a UV lamp (254 and 365 nm). Column chromatography was performed on silica gel (Silica 60, 0.015–0.04 mm, Macherey-Nagel, Duren, Germany). 4-Chloropyrimidines **3a** [53], **3b** [54], **3c** [55], **3d** [56] were obtained via described methods. All other starting materials were commercially available. All reagents except commercial products of satisfactory quality were purified according to literature procedures prior to use.

#### 3.1.2. S_N_Ar Reactions of 4-Chloropyrimidines **2a**–**d** with Amines (General Method) 

To the solution of chloropyrimidine **2a**–**d** (0.55 mmol) and corresponding amine (0.66 mmol) in isopropanol (5 mL), two drops of HCl were added. The reaction mixture was refluxed for 5 h. When chloropyrimidine **2d** was used as starting compound, reagent ratios and reaction time were modified, see below. Isopropanol was evaporated under reduced pressure and, to the residue, water (15 mL) was added and the resulting mixture was extracted with chloroform (3 × 15 mL). Combined organic layers were dried over MgSO_4_. The solvent was evaporated under reduced pressure and the product was isolated via preparative column chromatography.

*N*,2-Dimethyl-*N*-phenyl-5,6,7,8-tetrahydroquinazolin-4-amine (**2a**)

Yield 71 mg (51%). Yellowish solid, m.p. 84–85 °C; R_f_ = 0.61 (CHCl_3_–MeOH 10:1).

^1^H NMR (CDCl_3_, δ, ppm): 1.42–1.53 m (2H, CH_2_), 1.49–1.63 m (2H, CH_2_), 1.72–1.80 m (2H, CH_2_), 2.58 s (3H, CH_3_), 2.69–2.80 m (2H, CH_2_), 3.45 s (3H, CH_3_N), 6.95–7.03 m (2H, 2CH), 7.05–7.13 m (1H, CH), 7.24–7.34 m (2H, 2CH).

^13^C NMR (CDCl_3_, δ, ppm): 22.3 (CH_2_), 22.8 (CH_2_), 25.9 (CH_3_), 26.1 (CH_2_), 32.3 (CH_2_), 41.5 (CH_3_N), 116.2 (C(4a)), 124.2 (2CH), 124.4 (CH), 129.3 (2CH), 148.1 (C, Ph), 163.4 (C(4)), 163.9 (C(2)), 164.8 (C(8a)).

FT-IR (KBr tablet, ν, cm^−1^): 3440, 3055, 3034, 2932, 2862, 2657, 2436, 2360, 2335, 1976, 1938, 1910, 1865, 1834, 1734, 1596, 1558, 1495, 1468, 1438, 1423, 1384, 1333, 1278, 1218, 1186, 1158, 1119, 1100, 1074, 1028, 993, 904, 889, 861, 833, 759, 741, 709, 698, 662, 613, 580, 547, 497, 449.

MS (EI^+^, *m*/*z* (%)): 255 (1), 254 (14), 253 (87) [M]^+^, 252 (100), 238 (14) [M-CH_3_]^+^, 224 (11), 183 (5), 176 (11) [M-C_6_H_5_]^+^, 163 (5), 162 (55), 161 (12), 148 (8), 107 (7), 106 (20), 105 (5), 104 (7), 91 (6), 79 (11), 78 (5), 77 (21).

HRMS (ESI^+^, *m*/*z*): calculated for C_16_H_19_N_3_ [M + H]^+^ 254.1652, found 254.1651.

*N,2*-Dimethyl-*N*-(4-methylphenyl)-5,6,7,8-tetrahydroquinazolin-4-amine (**2b**) 

Yield 121 mg (82%). Yellowish solid, m.p. 91–92 °C; R_f_ = 0.38 (CHCl_3_–MeOH 10:1).

^1^H NMR (CDCl_3_, δ, ppm): 1.42–1.53 m (2H, CH_2_), 1.62–1.78 m (4H, 2CH_2_), 2.33 s (3H, CH_3_Ar), 2.57 s (3H, CH_3_), 2.67–2.76 br.t (2H, CH_2_, 3.41 c (3H, CH_3_N), 6.87–6.93 m (2H, 2CH), 7.06–7.12 m (2H, 2CH).

^13^C NMR (CDCl_3_, δ, ppm): 21.1 (CH_3_Ar), 22.4 (CH_2_), 22.8 (CH_2_), 25.9 (CH_3_), 26.1 (CH_2_), 32.4 (CH_2_), 41.8 (CH_3_N), 115.8 (C(4a)), 124.6 (2CH), 130.0 (2CH), 134.3 (C, Ar), 145.7 (C, Ar), 163.3 (C(4)), 163.8 (C(2)), 164.5 (C(8a)).

FT-IR (KBr tablet, ν, cm^−1^): 3434, 3020, 2969, 2929, 2859, 2651, 2435, 2370, 2329, 1949, 1925, 1816, 1729, 1559, 1542, 1512, 1472, 1447, 1412, 1390, 1282, 1257, 1216, 1182, 1157, 1118, 1102, 1069, 1023, 994, 949, 906, 892, 863, 829, 765, 734, 722, 654, 630, 597, 573, 542, 501, 473, 444.

MS (EI^+^, *m*/*z* (%)): 269 (1), 268 (17), 267 (93) [M]^+^, 266 (100), 252 (15) [M-CH_3_]^+^, 250 (5), 238 (9), 176 (9), 163 (6), 162 (59), 161 (10), 148 (7), 147 (5), 121 (5), 120 (20), 119 (5), 118 (6), 107 (11), 106 (7), 105 (7), 91 (13), 79 (10), 77 (10), 65 (5).

HRMS (ESI^+^, *m*/*z*): calculated for C_17_H_21_N_3_ [M + H]^+^ 268.1808, found 268.1811.

*N*-(4-methoxyphenyl)-*N*,2-dimethyl-5,6,7,8-tetrahydroquinazolin-4-amine (**2c**)

Yield 98 mg (63%). Beige solid, m.p. 87–88 °C; R_f_ = 0.68 (CHCl_3–_MeOH 10:1).

^1^H NMR (CDCl_3_, δ, ppm): 1.40–1.52 m (2H, CH_2_), 1.59–1.75 m (4H, 2CH_2_), 2.56 s (3H, CH_3_), 2.64–2.76 m (2H, CH_2_), 3.38 s (3H, CH_3_N), 3.80 s (3H, CH_3_O), 6.79–6.88 m (2H, 2CH), 6.92–7.02 m (2H, 2CH). 

^13^C NMR (CDCl_3_, δ, ppm): 22.3 (CH_2_), 22.9 (CH_2_), 25.9 (CH_3_), 26.1 (CH_2_), 32.3 (CH_2_), 42.2 (CH_3_N), 55.6 (CH_3_O), 114.5 (2CH), 115.2 (C(4a)), 126.6 (2CH), 141.5 (C, Ar), 156.9 (C-O, Ar), 163.5 (C(4)), 163.6 (C(2)), 164.3 (C(8a)).

FT-IR (KBr tablet, ν, cm^−1^): 3444, 3054, 3010, 2934, 2916, 2859, 2839, 2432, 2065, 1947, 1870, 1610, 1558, 1544, 1512, 1463, 1443, 1413, 1385, 1297, 1251, 1183, 1172, 1156, 1117, 1099, 1037, 993, 949, 904, 891, 841, 816, 768, 734, 656, 631, 596, 573, 553, 506, 468, 441.

MS (EI^+^, *m*/*z* (%)): 285 (2), 284 (17), 283 (100) [M]^+^, 282 (60), 268 (25) [M-CH_3_]^+^, 254 (5), 176 (6), 163 (8), 162 (69), 148 (9), 147 (11), 142 (5), 136 (13), 123 (9), 121 (19), 120 (8), 107 (8), 106 (8), 92 (5), 79 (15), 78 (6), 77 (15).

HRMS (ESI^+^, *m*/*z*): calculated for C_17_H_21_N_3_O [M + H]^+^ 284.1757, found 284.1757.

4-(3,4-Dihydroquinolin-1(2*H*)-yl)-2-methyl-5,6,7,8-tetrahydroquinazoline (**2d**)

Yield 108 mg (70%). Beige solid, m.p. 118–119 °C; R_f_ = 0.80 (CHCl_3–_MeOH 10:1).

^1^H NMR (CDCl_3_, δ, ppm): 1.48–1.65 m (2H, CH_2_, THQ), 1.72–1.85 m (2H, CH_2_, THQ), 1.96–2.13 m (4H, CH_2_, THQ + CH_2_, DHQ), 2.58 (3H, CH_3_), 2.72–2.90 m (4H, CH_2_, THQ + CH_2_, DHQ), 3.76–3.85 m (2H, CH_2_N, DHQ), 6.36–6.46 m (1H, CH), 6.80–6.88 m (1H, CH), 6.92–7.01 m (1H, CH), 7.04–7.13 m (1H, CH).

^13^C NMR (CDCl_3_, δ, ppm): 22.5 (CH_2_, THQ), 22.6 (CH_2_, THQ), 24.1 (CH_2_, DHQ), 25.8 (CH_3_), 25.9 (CH_2_, THQ), 27.1 (CH_2_, DHQ), 32.4 (CH_2_, THQ), 47.3 (CH_2_N, DHQ), 117.9 (CH), 118.7 (C(4a)), 121.2 (CH), 126.0 (CH), 127.8 (C, DHQ), 128.9 (CH), 141.7 (C, DHQ), 162.5 (C(4)), 164.7 (C(2)), 166.0 (C(8a)). 

FT-IR (KBr tablet, ν, cm^−1^): 3445, 3062, 3034, 2945, 2924, 2875, 2843, 2430, 1916, 1733, 1643, 1603, 1551, 1491, 1423, 1405, 1337, 1292, 1254, 1216, 1193, 1155, 1133, 1111, 1051, 1034, 994, 974, 945, 903, 886, 832, 759, 659, 622, 580, 547, 498, 426.

MS (EI^+^, *m*/*z* (%)): 281 (2), 280 (17), 279 (94) [M]^+^, 278 (100), 264 (10) [M-CH_3_]^+^, 251 (7), 250 (18), 237 (6), 236 (7), 209 (7), 162 (14), 149 (15), 148 (26), 147 (6), 133 (9), 132 (75), 131 (11), 130 (15), 126 (8), 117 (11), 115 (8), 107 (11), 106 (9), 104 (6), 91 (7), 80 (5), 79 (12), 78 (5), 77 (13).

HRMS (ESI^+^, *m*/*z*): calculated for C_18_H_21_N_3_ [M + H]^+^ 280.1808, found 280.1809. 

4-(6-Methoxy-3,4-dihydroquinolin-1(2*H*)-yl)-2-methyl-5,6,7,8-tetrahydroquinazoline (**2e**)

Yield 95 mg (56%). Yellowish solid, m.p. 116–117 °C; R_f_ = 0.46 (CHCl_3–_MeOH 20:1).

^1^H NMR (CDCl_3_, δ, ppm): 1.41–1.57 m (2H, CH_2_, THQ), 1.66–1.78 m (2H, CH_2_, THQ), 1.82–1.91 m (2H, CH_2_, THQ), 1.92–2.03 m (2H, CH_2_, DHQ), 2.57 s (3H, CH_3_), 2.67–2.77 m (2H, CH_2_, DHQ), 2.78–2.88 m (2H, CH_2_, THQ), 3.72 s (3H, CH_3_O), 3.75–3.85 m (2H, CH_2_N, DHQ), 6.45 d (1H, ^3^*J* = 8.7 Hz, CH), 6.57 dd (1H, ^3^*J* = 8.7 Hz, ^4^*J* = 2.7 Hz, CH), 6.65 d (1H, ^4^*J* = 2.7 Hz, CH). 

^13^C NMR (CDCl_3_, δ, ppm): 21.8 (CH_2_, THQ), 22.4 (CH_2_, THQ), 24.1 (CH_2_, DHQ), 24.6 (CH_3_), 26.3 (CH_2_, THQ), 27.0 (CH_2_, DHQ), 30.9 (CH_2_, THQ), 47.0 (CH_2_N, DHQ), 55.4 (CH_3_O), 111.2 (CH), 113.8 (CH), 116.7 (C(4a)), 120.6 (CH), 131.5 (C, DHQ), 134.3 (C, DHQ), 155.4 (C-O, DHQ), 162.1 (C(4)), 162.9 (C(2), C(8a)). 

FT-IR (KBr tablet, ν, cm^−1^): 3470, 3428, 3239, 3055, 2936, 2858, 2747, 2688, 2583, 2361,1777, 1614, 1552, 1502, 1467, 1438, 1404, 1320, 1296, 1262, 1236, 1204, 1163, 1136, 1114, 1054, 1038, 1024, 995, 931, 882, 864, 833, 781, 758, 738, 714, 655, 630, 604, 575, 548, 451, 429.

MS (EI^+^, *m*/*z* (%)): 311 (2), 310 (20), 309 (100) [M]^+^, 308 (55), 281 (5), 280 (9), 266 (7), 163 (9), 162 (73), 161 (11), 160 (6), 155 (8), 149 (9), 148 (20), 147 (20), 146 (5), 141 (5), 131 (5), 130 (5), 107 (8), 106 (9), 79 (19), 77 (13).

HRMS (ESI^+^, *m*/*z*): calculated for C_19_H_23_N_3_O [M + H]^+^ 310.1914, found 310.1916. 

2-Methyl-4-(7-nitro-3,4-dihydroquinolin-1(2*H*)-yl)-5,6,7,8-tetrahydroquinazoline (**2f**)

Yield 110 mg (62%). Yellow solid, m.p. 184–185 °C; R_f_ = 0.14 (light petrol–EtOAc–MeOH 3:1:0.01).

^1^H NMR (CDCl_3_, δ, ppm): 1.59–1.72 m (2H, CH_2_, THQ), 1.82–1.91 m (2H, CH_2_, THQ), 2.05–2.14 m (2H, CH_2_, DHQ), 2.14–2.22 m (2H, CH_2_, THQ), 2.62 s (3H, CH_3_), 2.84–2.98 m (4H, CH_2_, THQ + CH_2_, DHQ), 3.76–3.85 m (2H, CH_2_N, DHQ), 7.18 d (1H, ^4^*J* = 2.2 Hz, CH), 7.20 d (1H, ^3^*J* = 8.3 Hz, CH), 7.65 dd (1H, ^3^*J* = 8.3 Hz, ^4^*J* = 2.2 Hz, CH). 

^13^C NMR (CDCl_3_, δ, ppm): 22.44 (CH_2_, THQ), 22.46 (CH_2_, THQ), 22.8 (CH_2_, DHQ), 25.7 (CH_2_, THQ), 25.8 (CH_3_), 27.7 (CH_2_, DHQ), 32.5 (CH_2_, THQ), 47.7 (CH_2_, DHQ), 111.3 (CH), 114.9 (CH), 119.9 (C(4a)), 129.8 (CH), 133.3 (C, DHQ), 142.5 (C, DHQ), 146.9 (CNO_2_, DHQ), 162.1 (C(4)), 165.5 (C(2)), 168.0 (C(8a)). 

FT-IR (KBr tablet, ν, cm^−1^): 3446, 2949, 2911, 2862, 2423, 1919, 1775, 1617, 1550, 1518, 1494, 1458, 1421, 1394, 1342, 1275, 1248, 1213, 1154, 1135, 1119, 1097, 1050, 996, 923, 884, 862, 821, 803, 758, 736, 683, 659, 627, 584, 523, 475.

MS (EI^+^, *m*/*z* (%)): 326 (1), 325 (12), 324 (57) [M]^+^, 323 (12), 309 (6) [M-CH_3_]^+^, 308 (6), 307 (26), 296 (5), 290 (12), 282 (6), 277 (5), 249 (5), 177 (22), 173 (5), 163 (5), 162 (22), 160 (6), 149 (19), 148 (100), 147 (13), 131 (7), 130 (10), 121 (7), 107 (31), 106 (15), 104 (9), 103 (5), 80 (9), 79 (17), 78 (7), 77 (17), 42 (5).

HRMS (ESI^+^, *m*/*z*): calculated for C_18_H_20_N_4_O_2_ [M + H]^+^ 325.1659, found 325.1655. 

*N*^4^-Methyl-*N*^4^-phenyl-5,6,7,8-tetrahydroquinazolinyl-2,4-diamine (**2g**)

Yield 71 mg (51%). Beige solid, m.p. 128–129 °C; R_f_ = 0.38 (CHCl_3–_MeOH 10:1).

^1^H NMR (CDCl_3_, δ, ppm): 1.35–1.48 m (2H, CH_2_), 1.53–1.69 m (4H, 2CH_2_), 2.57–2.71 m (2H, CH_2_), 3.39 s (3H, CH_3_), 5.31 br.s (2H, NH_2_), 6.98–7.07 m (2H, 2CH), 7.12–7.21 m (1H, CH), 7.27–7.36 m (2H, 2CH). 

^13^C NMR (CDCl_3_, δ, ppm): 21.8 (CH_2_), 23.0 (CH_2_), 25.7 (CH_2_), 30.9 (CH_2_), 41.9 (CH_3_N), 109.3 (C(4a)), 125.0 (2CH), 125.2 (CH), 129.4 (2CH), 147.3 (C, Ph), 158.6 (C(2)), 162.0 (C(8a)), 164.5 (C(4)). 

HRMS (ESI^+^, *m*/*z*): calculated for C_15_H_18_N_4_ [M + H]^+^ 255.1604, found 255.1599. 

*N*^4^-Methyl-*N*^4^-(4-methylphenyl)-5,6,7,8-tetrahydroquinazolinyl-2,4-diamine (**2h**)

Yield 74 mg (50%). Beige solid, m.p. 135–136 °C; R_f_ = 0.31 (CHCl_3–_MeOH 10:1).

^1^H NMR (CDCl_3_, δ, ppm): 1.30–1.45 m (2H, CH_2_), 1.45–1.54 m (2H, CH_2_), 1.54–1.64 m (2H, CH_2_), 2.35 s (3H, CH_3_), 2.62–2.74 m (2H, CH_2_), 3.39 s (3H, CH_3_N), 6.24 br.s (2H, NH_2_), 6.91–7.03 m (2H, 2CH), 7.11–7.20 m (2H, 2CH). 

^13^C NMR (CDCl_3_, δ, ppm): 20.7 (CH_2_), 21.1 (CH_3_, Ar), 22.6 (CH_2_), 25.4 (CH_2_), 28.4 (CH_2_), 42.9 (CH_3_N), 108.0 (C(4a)), 126.0 (2CH), 130.2 (2CH), 137.1 (C, Ar), 143.1 (C, Ar), 155.16 (C(2)), 155.21 (C(8a)), 164.3 (C(4)). 

HRMS (ESI^+^, *m*/*z*): calculated for C_16_H_20_N_4_ [M + H]^+^ 269.1761, found 269.1755. 

*N*^4^-(4-Methoxyphenyl)-*N*^4^-methyl-5,6,7,8-tetrahydroquinazolinyl-2,4-diamine (**2i**)

Yield 59 mg (38%). Beige solid, m.p. 132–133 °C; R_f_ = 0.4 (CHCl_3–_MeOH 10:1).

^1^H NMR (CDCl_3_, δ, ppm): 1.39–1.48 m (2H, CH_2_), 1.57–1.67 m (4H, 2CH_2_), 2.54–2.62 m (2H, CH_2_), 3.31 s (3H, CH_3_N), 3.81 s (3H, CH_3_O), 4.69 br.s (2H, NH_2_), 6.78–6.86 m (2H, 2CH), 6.95–7.01 m (2H, 2CH). 

^13^C NMR (CDCl_3_, δ, ppm): 22.4 (CH_2_), 23.2 (CH_2_), 25.9 (CH_2_), 32.3 (CH_2_), 42.2 (CH_3_N), 55.6 (CH_3_O), 109.0 (C(4a)), 114.5 (2CH), 126.7 (2CH), 141.5 (C, Ar), 156.8 (C, Ar), 160.1 (C), 164.7 (C), 164.9 (C). 

HRMS (ESI^+^, *m*/*z*): calculated for C_16_H_20_N_4_O [M + H]^+^ 285.1710, found 285.1707. 

2-Amino-4-(6-methoxy-3,4-dihydroquinolin-1(2*H*)-yl)-5,6,7,8-tetrahydroquinazoline (**2j**)

Yield 61 mg (36%). Beige solid, m.p. 145–146 °C; R_f_ = 0.65 (light petrol–EtOAc 2:3).

^1^H NMR (CDCl_3_, δ, ppm): 1.44–1.59 m (2H, CH_2_, THQ), 1.67–1.80 m (2H, CH_2_, THQ), 1.83–1.94 m (2H, CH_2_, THQ), 1.95–2.06 m (2H, CH_2_, DHQ), 2.60–2.72 m (2H, CH_2_, THQ), 2.72–2.81 m (2H, CH_2_, DHQ), 3.66–3.80 m (2H, CH_2_, DHQ), 3.78 s (3H, CH_3_O), 5.06 br.s (2H, NH_2_), 6.55 d (1H, ^3^*J* = 8.7 Hz, CH), 6.62 dd (1H, ^3^*J* = 8.7 Hz, ^4^*J* = 2.7 Hz, CH), 6.70 d (1H, ^4^*J* = 2.7 Hz, CH). 

^13^C NMR (CDCl_3_, δ, ppm): 22.3 (CH_2_), 23.0 (CH_2_), 24.2 (CH_2_), 26.1 (CH_2_), 27.3 (CH_2_), 31.6 (CH_2_), 47.0 (CH_2_N), 55.6 (CH_3_O), 111.0 (C(4a), THQ), 111.5 (CH), 113.8 (CH), 121.0 (CH), 131.0 (C, DHQ), 135.0 (C, DHQ), 155.1 (C, DHQ), 155.1 (C, THQ), 159.8 (C, THQ), 163.7 (C, THQ). 

HRMS (ESI^+^, *m*/*z*): calculated for C_18_H_22_N_4_O [M + H]^+^ 311.1866, found 311.1863. 

*N*-(4-Methoxyphenyl)-*N*,2-dimethyl-6,7,8,9-tetrahydro-5*H*-cyclohepta[*d*]pyrimidin-4-amine (**2k**)

Yield 75 mg (46%). Beige solid, m.p. 92–93 °C; R_f_ = 0.66 (CHCl_3–_MeOH 10:1).

^1^H NMR (CDCl_3_, δ, ppm): 1.07–1.18 m (2H, CH_2_), 1.53–1.71 m (4H, 2CH_2_), 2.16–2.29 m (2H, CH_2_), 2.58 s (3H, CH_3_), 2.76–2.88 m (2H, CH_2_), 3.35 s (3H, CH_3_N), 3.78 s (3H, CH_3_O), 6.77–6.85 m (2H, 2CH), 6.87–6.96 m (2H, 2CH). 

^13^C NMR (CDCl_3_, δ, ppm): 25.7 (CH_3_), 25.8 (CH_2_), 26.3 (CH_2_), 27.7 (CH_2_), 31.9 (CH_2_), 38.6 (CH_2_), 42.1 (CH_3_N), 55.6 (CH_3_O), 114.8 (2CH, Ar), 120.5 (C(4a)), 125.1 (2CH, Ar), 142.9 (C, Ar), 156.5 (C, Ar), 163.4 (C(2)), 163.5 (C(4)), 171.0 (C(9a)). 

HRMS (ESI^+^, *m*/*z*): calculated for C_18_H_23_N_3_O [M + H]^+^ 298.1914, found 298.1917. 

4-(6-Methoxy-3,4-dihydroquinolin-1(2*H*)-yl)-2-methyl-6,7,8,9-tetrahydro-5*H*-cyclohepta[*d*]pyrimidine (**2l**)

Yield 107 mg (60%). Orange solid, m.p. 122–123 °C; R_f_ = 0.58 (CHCl_3–_MeOH 10:1).

^1^H NMR (CDCl_3_, δ, ppm): 1.37–1.50 m (2H, CH_2_), 1.65–1.84 m (4H, 2CH_2_), 1.98–2.03 m (2H, CH_2_, DHQ), 2.31–2.43 m (2H, CH_2_), 2.60 s (3H, CH_3_), 2.77–2.86 m (2H, CH_2_, DHQ), 2.90–3.00 m (2H, CH_2_), 3.69–3.82 m (2H, CH_2_N, DHQ), 3.76 s (3H, CH_3_O), 6.44 d (1H, ^3^*J* = 8.8 Hz, CH), 6.57 dd (1H, ^3^*J* = 8.8 Hz, ^4^*J* = 2.8 Hz, CH), 6.68 d (1H, ^4^*J* = 2.8 Hz, CH). 

^13^C NMR (CDCl_3_, δ, ppm): 23.7 (CH_2_, DHQ), 25.3 (CH_3_), 25.8 (CH_2_), 27.2 (CH_2_), 27.5 (CH_2_, DHQ), 28.2 (CH_2_), 32.1 (CH_2_), 38.4 (CH_2_), 48.0 (CH_2_N, DHQ), 55.6 (CH_3_O), 112.0 (CH), 114.3 (CH), 118.6 (CH), 123.1 (C(4a)), 128.7 (C, Ar), 136.4 (C, Ar), 154.5 (C-O, Ar), 162.4 (C), 164.1 (2C). 

HRMS (ESI^+^, *m*/*z*): calculated for C_20_H_25_N_3_O [M + H]^+^ 324.2070, found 324.2068. 

2-Chloro-*N*-(4-methoxyphenyl)-*N*-methyl-5,6,7,8-tetrahydroquinazolin-4-amine (**2m**)

Yield 27 mg (16%) from chloropyrimidine **2d** (0.83 mmol, 168 mg) and 4-methoxy-*N*-methylaniline (0.55 mmol, 75 mg), the reaction mixture was refluxed for 6 h. White solid, m.p. 138–139 °C; R_f_ = 0.82 (CHCl_3–_MeOH 20:1).

^1^H NMR (CDCl_3_, δ, ppm): 1.38–1.51 m (2H, CH_2_), 1.58–1.71 m (4H, 2CH_2_), 2.67–2.76 m (2H, CH_2_), 3.38 s (3H, CH_3_N), 3.81 s (3H, CH_3_O), 6.79–6.90 m (2H, 2CH), 6.97–7.03 m (2H, 2CH). 

^13^C NMR (CDCl_3_, δ, ppm): 21.9 (CH_2_), 22.6 (CH_2_), 26.2 (CH_2_), 32.4 (CH_2_), 42.7 (CH_3_N), 55.6 (CH_3_O), 114.7 (2CH), 116.1 (C(4a)), 127.1 (2CH), 140.2 (C, Ar), 156.9 (C(2)), 157.7 (C, Ar), 164.7 (C(4)), 167.1 (C(8a)). 

HRMS (ESI^+^, *m*/*z*): calculated for C_16_H_18_ClN_3_O [M + H]^+^ 304.1211, 306.1183; found 304.1206, 306.1181. 

*N*,*N′*-bis(4-methoxyphenyl)-*N,N′*-dimethyl-5,6,7,8-tetrahydroquinazolin-2,4-diamine (**2n**)

Yield 17 mg (13%), from chloropyrimidine **2d** (0.55 mmol, 112 mg) and 4-methoxy-*N*-methylaniline (0.66 mmol, 90 mg), the reaction mixture was refluxed for 12 h. Brown amorphous solid, m.p. 35–40 °C; R_f_ = 0.18 (CH_2_Cl_2–_MeOH 20:1).

^1^H NMR (CDCl_3_, δ, ppm): 1.35–1.50 m (2H, CH_2_), 1.55–1.73 m (4H, 2CH_2_), 2.55–2.67 m (2H, CH_2_), 3.17 s (3H, CH_3_N), 3.53 s (3H, CH_3_N), 3.80 s (3H, CH_3_O), 3.82 s (3H, CH_3_O), 6.78–6.85 m (2H, 2CH), 6.84–6.93 m (2H, 2CH), 6.93–7.00 m (2H, 2CH), 7.28–7.35 m (2H, 2CH). 

^13^C NMR (CDCl_3_, δ, ppm): 22.6 (CH_2_), 23.4 (CH_2_), 25.9 (CH_2_), 32.9 (CH_2_), 38.4 (CH_3_N), 42.7 (CH_3_N), 55.54 (CH_3_O), 55.58 (CH_3_O), 108.0 (C(4a)), 113.6 (2CH), 114.3 (2CH), 126.7 (2CH), 127.4 (2CH), 139.8 (C, Ar), 141.8 (C, Ar), 156.3 (C, Ar), 156.6 (C, Ar), 159.5 (C(2)), 163.6 (C(4)), 165.0 (C(8a)). 

HRMS (ESI^+^, *m*/*z*): calculated for C_24_H_28_N_4_O_2_ [M + H]^+^ 405.2285, found 405.2284. 

2-Hydroxy-4-(6-methoxy-3,4-dihydroquinolin-1(2*H*)-yl)-5,6,7,8-tetrahydroquinazoline (**2o**)

Yield 18 mg (12%), from chloropyrimidine **2d** (0.96 mmol, 194 mg) and 4-methoxy-*N*-methylaniline (0.48 mmol, 78 mg), the reaction mixture was refluxed for 60 h. Yellowish solid, m.p. 185–187 °C; R_f_ = 0.67 (CHCl_3–_MeOH 20:1).

^1^H NMR (CDCl_3_, δ, ppm): 1.66–1.83 m (4H, 2CH_2_, THQ), 1.89–2.02 m (2H, CH_2_, DHQ), 2.37–2.49 m (2H, CH_2_, THQ), 2.49–2.60 m (2H, CH_2_, THQ), 2.67–2.79 m (2H, CH_2_, DHQ), 3.81 s (3H, CH_3_), 3.88–3.97 m (2H, CH_2_, DHQ), 6.69–6.84 m (2H, 2CH), 7.07–7.15 m (1H, CH), 8.73 br.s (1H, OH). 

^13^C NMR (CDCl_3_, δ, ppm): 21.8 (CH_2_, THQ), 22.4 (CH_2_, THQ), 22.7 (CH_2_, THQ), 23.8 (CH_2_, DHQ), 27.1 (CH_2_, DHQ), 32.5 (CH_2_, THQ), 45.1 (CH_2_N, DHQ), 55.7 (CH_3_O), 111.9 (C(4a)), 112.9 (CH), 115.5 (CH), 123.4 (CH), 130.1 (C, Ar), 135.2 (C, Ar), 150.0 (C(4)), 157.4 (C, Ar), 162.89 (C), 162.93 (C). 

HRMS (ESI^+^, *m*/*z*): calculated for C_18_H_21_N_3_O_2_ [M + H]^+^ 312.1707, found 312.1695. 

2,4-Bis(6-methoxy-3,4-dihydroquinolin-1(2*H*)-yl)-5,6,7,8-tetrahydroquinazoline (**2p**)

Yield 6 mg (6%), from chloropyrimidine **2d** (0.96 mmol, 194 mg) and 4-methoxy-*N*-methylaniline (0.48 mmol, 78 mg), the reaction mixture was refluxed for 60 h. Yellow solid, m.p. 150–152 °C; R_f_ = 0.46 (light petrol–EtOAc 4:1).

^1^H NMR (CDCl_3_, δ, ppm): 1.49–1.61 m (2H, CH_2_, THQ), 1.71–1.82 m (2H, CH_2_, THQ), 1.88–2.08 m (6H, CH_2_, THQ + 2CH_2_, DHQ), 2.64–2.88 m (6H, CH_2_, THQ + 2CH_2_, DHQ), 3.68–3.75 m (2H, CH_2_, DHQ), 3.77 s (3H, CH_3_), 3.78 s (3H, CH_3_), 3.97–4.10 m (2H, CH_2_, DHQ), 6.51 d (1H, ^3^*J* = 8.8 Hz, CH), 6.59 dd (1H, ^3^*J* = 8.8 Hz, ^4^*J* = 2.8 Hz, CH), 6.62–6.73 m (3H, 3CH), 7.7d0 d (1H, ^3^*J* = 9.0 Hz, CH). 

^13^C NMR (CDCl_3_, δ, ppm): 22.9 (CH_2_, THQ), 23.2 (CH_2_, THQ), 24.0 (CH_2_, DHQ), 24.5 (CH_2_, DHQ), 26.2 (CH_2_, THQ), 27.5 (CH_2_, DHQ), 28.2 (CH_2_, DHQ), 32.9 (CH_2_, THQ), 45.0 (CH_2_N, DHQ), 46.7 (CH_2_N, DHQ), 55.6 (2CH_3_O), 111.1 (C(4a)), 111.3 (CH), 111.4 (CH), 113.0 (CH), 113.6 (CH), 120.3 (CH), 125.6 (CH), 130.7 (C, Ar), 131.1 (C, Ar), 133.9 (C, Ar), 135.7 (C, Ar), 154.4 (C-O, Ar), 154.7 (C-O, Ar), 159.1 (C(2)), 162.5 (C(4)), 166.1 (C(8a)). 

HRMS (ESI^+^, *m*/*z*): calculated for C_28_H_32_N_4_O_2_ [M + H]^+^ 457.2598, found 457.2586. 

2-Isopropoxy-4-(6-methoxy-3,4-dihydroquinolin-1(2H)-yl)-5,6,7,8-tetrahydroquinazoline (**2q**)

Yield 4 mg (2%), from chloropyrimidine **2d** (0.96 mmol, 194 mg) and 4-methoxy-*N*-methylaniline (0.48 mmol, 78 mg), the reaction mixture was refluxed for 60 h. Yellowish solid, m.p. 135–137 °C; R_f_ = 0.33 (light petrol–EtOAc 4:1).

^1^H NMR (CDCl_3_, δ, ppm): 1.37 d (6H, ^3^*J* = 6.2 Hz, 2CH_3_, *i*-Pr), 1.49–1.57 m (2H, CH_2_, THQ), 1.71–1.80 m (2H, CH_2_, THQ), 1.88–1.96 m (2H, CH_2_, THQ), 1.96–2.07 m (2H, CH_2_, DHQ), 2.70–2.80 m (4H, CH_2_, THQ + CH_2_, DHQ), 3.78 s (3H, CH_3_), 3.75–3.83 m (2H, CH_2_, DHQ), 5.23 quint (1H, ^3^*J* = 6.2 Hz, CH, *i*-Pr), 6.51 d (1H, ^3^*J* = 8.8 Hz, CH), 6.61 dd (1H, ^3^*J* = 8.8 Hz, ^4^*J* = 2.9 Hz, CH), 6.70 d (1H, ^4^*J* = 2.9 Hz, CH). 

^13^C NMR (CDCl_3_, δ, ppm): 22.2 (2CH_3_, *i*-Pr), 22.6 (CH_2_, THQ), 23.0 (CH_2_, THQ), 24.4 (CH_2_, DHQ), 26.3 (CH_2_, THQ), 27.4 (CH_2_, DHQ), 32.7 (CH_2_, THQ), 46.7 (CH_2_, DHQ), 55.6 (CH_3_O), 69.1 (CH-O, *i*-Pr), 111.3 (CH), 113.1 (C(4a)), 113.7 (CH), 120.5 (CH), 131.3 (C, Ar), 135.3 (C, Ar), 154.9 (C-O, Ar), 162.7 (C), 163.9 (C), 167.6 (C). 

HRMS (ESI^+^, *m*/*z*): calculated for C_21_H_27_N_3_O_2_ [M + H]^+^ 354.2176, found 354.2166. 

4-Isopropoxy-2-(6-methoxy-3,4-dihydroquinolin-1(2H)-yl)-5,6,7,8-tetrahydroquinazoline (**2r**)

Yield 34 mg (20%), from chloropyrimidine **2d** (0.96 mmol, 194 mg) and 4-methoxy-*N*-methylaniline (0.48 mmol, 78 mg), the reaction mixture was refluxed for 60 h. White solid, m.p. 94–95 °C; R_f_ = 0.75 (light petrol–EtOAc 2:1).

^1^H NMR (CDCl_3_, δ, ppm): 1.30 d (6H, ^3^*J* = 6.2 Hz, 2CH_3_, *i*-Pr), 1.69–1.83 m (4H, 2CH_2_, THQ), 1.88–2.01 m (2H, CH_2_, DHQ), 2.36–2.49 m (2H, CH_2_, THQ), 2.57–2.69 m (2H, CH_2_, THQ), 2.70–2.80 m (2H, CH_2_, DHQ), 3.79 s (3H, CH_3_), 3.93–4.04 m (2H, CH_2_, DHQ), 5.22 quint (1H, ^3^*J* = 6.2 Hz, CH-O, *i*-Pr), 6.64 d (1H, ^4^*J* = 2.9 Hz, CH), 6.70 dd (1H, ^3^*J* = 9.0 Hz, ^4^*J* = 2.9 Hz, CH), 7.73 d (1H, ^3^*J* = 9.0 Hz, CH). 

^13^C NMR (CDCl_3_, δ, ppm): 21.6 (CH_2_, THQ), 22.2 (2CH_3_, *i*-Pr), 22.7 (CH_2_, THQ), 22.8 (CH_2_, THQ), 24.0 (CH_2_, DHQ), 28.1 (CH_2_, DHQ), 32.4 (CH_2_, THQ), 45.1 (CH_2_, DHQ), 55.6 (CH_3_O), 68.2 (CH-O, *i*-Pr), 106.9 (C(4a)), 111.3 (CH), 112.8 (CH), 125.9 (CH), 131.2 (C, Ar), 133.9 (C, Ar), 154.7 (C-O, Ar), 158.5 (C(2)), 165.2 (C(8a)), 166.9 (C(4)). 

HRMS (ESI^+^, *m*/*z*): calculated for C_21_H_27_N_3_O_2_ [M + H]^+^ 354.2176, found 354.2169.

### 3.2. Biology

#### 3.2.1. Cell Cultures

Human breast cancer cell line MCF7′ and human lung adenocarcinoma cell line A549 were kindly provided by Dr. S. Dmitriev (Lomonosov Moscow State University), immortalized human fibroblast cell line VA13 (WI38 subline 2RA) was kindly provided by Dr. M. Rubtsova (Lomonosov Moscow State University), human embryonic kidney HEK293T cell line was kindly provided by Dr. E. Knyazhanskaya (Lomonosov Moscow State University). MCF7′, VA13, A549 and HEK293T cell lines were maintained in DMEM/F-12 (Thermo Fisher Scientific, Waltham, MA, USA) culture medium containing 10% fetal bovine serum (Thermo Fisher Scientific, Waltham, MA, USA) and 50 μg/mL penicillin and 0.05 mg/mL streptomycin at 37 °C (Thermo Fisher Scientific, Waltham, MA, USA) in 5% CO_2_. Cells were maintained at 37 °C in a humidified MCO-18AC incubator (Sanyo, Tokyo, Japan) supplied with 5% CO_2_. Cell cultures were tested for the absence of mycoplasma and validated by STR.

#### 3.2.2. Cell Viability Assay (MTT Assay)

The cytotoxicity of the substances was tested using the 3-(4,5-dimethylthiazol-2-yl)2,5-diphenyl tetrazolium bromide (MTT) assay [41] with some modifications. A total of 2500 cells per well for the MCF7′, HEK293T, A549 cell lines or 4000 cells per well for the VA-13 cell line were plated out in 135 μL of DMEM-F12 media (Gibco, Thermo Fisher Scientific, Waltham, MA, USA) in a 96-well plate and incubated in the 5% CO_2_ incubator for the first 20 h without treatment. Then, 15.8 μL of media–DMSO solutions of tested substances (final DMSO concentrations in the media were 0.5% or less) was added to the cells (triplicate each) and cells were treated for 64 h with 0.023 mg/L–50 mg/L (eight dilutions) of compounds **2f**,**n**,**o**,**r**; 1.14 μg/L–2.5 mg/L (eight dilutions) of compounds **2a**,**b**,**d**,**g**,**h**,**k**,**l**,**p**,**q**; 57.2 ng/L–125 μg/L (eight dilutions) of compounds **2e**,**i**,**j**; 22.9 ng/L–50 μg/L (eight dilutions) of compounds **1**,**2c**,**m**. Doxorubicin was used as a control substance. Then, the MTT reagent (Paneco LLC, Moscow, Russia) was added to cells up to a final concentration of 0.5 g/L (10 × stock solution in PBS was used) and incubated for 2 h at 37 °C in the incubator, under an atmosphere of 5% CO_2_. Then, the MTT solution was discarded, and 140 μL of DMSO (PharmaMed LLC, Moscow, Russia) was added. The plates were swayed on a shaker (120 rpm) to dissolve the formazan. The absorbance was measured using a microplate reader (VICTOR X5 Light Plate Reader, PerkinElmer, Waltham, MA, USA) at a wavelength of 555 nm (in order to measure formazan concentration). The results were used to construct dose–response graphs and to estimate IC_50_ values (IC_50_ is the concentration resulting in half of the maximal cytotoxic effect) with GraphPad Prism V, GraphPadSoftware, Inc., San Diego, CA, USA.

#### 3.2.3. Immunofluorescent Microscopy

A549 cells were seeded on coverslips (treated with poly-l-lysine at a concentration of 33 mg/L for an hour). Compounds were added up to the corresponding concentration after 24 h attachment and growth of the cells. Then, cells were incubated 24 h.

The cells were fixed with methanol (precooled in liquid nitrogen for 20 min), no additional permeabilization was performed. To prevent non-specific binding, cells were incubated with 4% BSA in PBS for one hour. Then, a solution of mouse monoclonal antibodies against alpha-tubulin cross-linked with Alexa 488 fluoroform (Thermo Fisher Scientific, Waltham, MA, USA, 32-2588) at a concentration of 5 μg/mL in 1% BSA solution in PBS was added to the cells, and the slides were incubated at +5 °C in a water chamber for 24 h. After the end of the incubation time, the cells were washed several times with PBS buffer solution and fixed on a glass slide with Moviol. Next day, cells were imaged with a ZEISS LSM 900 (ZEISS Microscopy, GmbH, Germany) and images processed with ImageJ.

#### 3.2.4. Tubulin + MAP Polymerization 

A tubulin + MAP polymerization assay was conducted as previously described in [57].

#### 3.2.5. Cell Death

HCT116 (colon carcinoma) cells (3 × 10^5^ cells in 2 mL of DMEM) were seeded in a 6-well plate and incubated with compounds **1**,**2c**,**e**,**i**,**j**,**m** and cisplatin at 2 × IC_50_ (values based on MTT assay, see Appendix A) for 48 h. After incubation, the cells were harvested by trypsinization, precipitated by centrifugation (3000 rpm), washed with cold PBS and recentrifuged. Aliquots of cells were processed as recommended in the Muse Annexin V& Dead Cell Kit (Luminex Corporation, Austin, TX, USA). The results were recorded on a Muse Cell Analyzer flow cytometer (Merck, Rahway, NJ, USA).

### 3.3. MTT Study of Compound **2c**, Encapsulated in Alginate-Based Nanocontainers

#### 3.3.1. General Remarks

The following reagents were used: sodium alginate (Na-Alg), calcium chloride (CaCl_2_), magnesium sulfate (MgSO_4_); all reagents from Sigma (Merck, Rahway, NJ, USA). 

Dynamic light scattering (DLS) measurements were carried out in 0.15 M NaCl aqueous solution using a Photocor Complex photometer (Photocor Instruments, Moscow, Russia) equipped with a He-Ne 10 mW laser (*λ* = 633 nm) as the light source. The data were processed using DynaLS (DynaLS Version 2.7.1) software. Mean hydrodynamic sizes of particles were determined by DLS at the fixed scattering angle (90°). Software provided by the manufacturer was employed to calculate diameter values. Electrophoretic mobility (EPM) of particles was measured by laser microelectrophoresis in a thermostatic cell using a Brookhaven Zeta Plus (Brookhaven Instruments, Holtsville, NY, USA) device with the corresponding software. UV/vis spectroscopy measurements were performed with a UV-PC PE-5400UV spectrophotometer (Ecros, Moscow, Russia).

#### 3.3.2. Preparation of Nanocontainers

All nanocontainers were synthesized at room temperature. A dilute solution of CaCl_2_ (1 mL) containing 0.56 mg of the compound was added dropwise to 50 mL of a 0.1% Na-Alg solution. A dilute solution of MgSO_4_ (1 mL) containing 0.60 mg of the compound was added dropwise to 50 mL of a 0.1% Na-Alg solution.

The resulting mixtures was intensely stirred for 24 h and placed in dialysis bags (MWCO~12 kDa, «Sigma», Merck, Rahway, NJ, USA) and dialyzed for 24 h against deionized water. After dialysis, the solutions were lyophilized.

#### 3.3.3. Preparation of Nanocontainers Filled with **2c**

A solution containing 1 mg of aminopyrimidine in 50 μL HCl was added dropwise to 5 mL of an aqueous solution of the Alg–Ca^2+^ (and Alg–Mg^2+^) nanocontainer containing 5 mg of the compound. The reaction mixture was intensely stirred for 24 h and neutralized by NaOH solution to pH~7. High-molecular products were purified by flow dialysis. Then, lyophilization was used.

#### 3.3.4. Cytotoxicity Studies of Nanocontainers Filled with **2c**

Lung adenocarcinoma A549, breast adenocarcinoma MDA-MB-231 and non-tumor fibroblasts immortalized with h-TERT cell lines were obtained from the American Type Culture Collection (Manassas, VA, USA). Cells were routinely propagated in Dulbecco’s modified Eagle’s medium (DMEM) supplemented with 10% fetal bovine serum (HyClone, Logan, UT, USA), 2 mM L-glutamine, 100 U/mL penicillin and 100 μg/mL streptomycin (PanEco, Moscow, Russia) at 37 °C, 5% CO_2_ in a humidified atmosphere. Cells in the logarithmic phase of growth were used in the experiments.

The cytotoxicity of compounds for human cell lines (A549, MDA-MB-231, non-tumor fibroblasts) was assessed in MTT assays. Cells were plated in 96-well plates (NUNC, Thermo Fisher Scientific, Waltham, MA, USA, 5∙10^3^ cells in 190 μL culture medium per well). After 24 h at 37 °C, 5% CO_2_ in a humidified atmosphere, cells were treated with compounds dissolved with DMEM from stocks: **2c**∙HCl–(Alg)–Ca^2+^ (C**_2c_** 100 μM, C_COO-groups_ 1000 μM), (Alg)–Ca^2+^ (C_COO-groups_ 1000 μM), **2c**∙HCl–(Alg)–Mg^2+^ (C**_2c_** 100 μM, C_COO-groups_ 1000 μM), (Alg)–Mg^2+^ (C_COO-groups_ 1000 μM), **2c** (C**_2c_** 200μM in DMSO) to final amine concentrations of 0.8–25 μM or equivalent amounts of samples (Alg)–Ca^2+^ and (Alg)–Mg^2+^ without amine. Doxorubicin (Teva, Amsterdam, Netherlands) was used as control compound. 

Cytotoxicity was assessed in a formazan conversion assay (MTT test) after a 48 h exposure. After the completion of drug exposure, 0.5 mg/mL 3-(4,5-dimethylthiazol-2-yl)-2,5-diphenyltetrazolium bromide (MTT reagent) was added to cells for 2 h, the culture medium was removed, cells were resuspended in 100 μL DMSO and the optical densities were measured on a Multiscan FC plate spectrophotometer (Thermo Fisher Scientific, Waltham, MA, USA) at a wavelength of 571 nm. The percentage of surviving cells for each dose was calculated as the quotient of the average optical density in the wells after incubation with this dose to the average optical density of the control wells (the values of the latter are taken as 100%). Five independent experiments were performed for each concentration. Standard deviations did not exceed 10%.

### 3.4. Molecular Dynamics Simulation

Molecular dynamics simulation was performed according to a previously published detailed protocol [57] using the model obtained from the tetramer structure of α,β-tubulin dimer PDB ID: 6GJ4 [58] and CHARMM36/CGenFF 4.4 force field [59,60] in GROMACS 2021.2 software [61] (the starting structure of the protein–ligand complex was obtained by means of molecular docking using AutoDock Vina 1.1.2 software [62]). For the visualization of the results, CPPTRAJ software 5.1.0 [63] in the Amber-Tools 21 package [64] and UCSF Chimera software 1.16 [65] were used; the position of verubulin (PDB ID: 5XKF [46]) was matched to the β-subunit for comparison.

## 4. Conclusions

In summary, a series of verubulin analogues, where the quinazoline core was replaced by tetrahydroquinazoline or cyclohepta[*d*]pyrimidine, with varying substituents at positions 2 and 4 of pyrimidine were obtained. The estimation of their cytotoxicity via MTT assay showed a spectrum of cytotoxic action from close to that of verubuliN′s to almost non-toxicity, offering useful information for SAR analysis and further research. 2-Methyl- and 2-chlorotetrahydroquinazolines, containing 4-methoxy-*N*-methylaniline moiety at position 4, were found to be the most potent: they revealed cytotoxicity with IC_50_ down to 1–4 nM, depending on the cancer (or non-cancer) cell line. According to fluorescent microscopy data, compounds that showed cytotoxicity in the MTT assay disrupt the normal cytoskeleton of the cell, and the pattern of destruction is similar to that for combretastatin A-4. Encapsulation of 4-aminotetrahydroquinazoline **2c** into alginate-based nanocontainers was performed to show that the cytotoxic action of the encapsulated compound **2c** is preserved compared to its free form, that opens the way to the further search for drug delivery systems for verubulin analogues. 

## Data Availability

Data is contained within the article and Appendix A.

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
