# Peer review of "Verubulin (Azixa) Analogues with Increased Saturation: Synthesis, SAR and Encapsulation in Biocompatible Nanocontainers Based on Ca2+ or Mg2+ Cross-Linked Alginate"

_pharmaceuticals, 2023, doi:10.3390/ph16101499_

Round 1
Reviewer 1 Report
In this study, a series of Verubulin analogues were obtained and their cytotoxicity towards cancer and non-cancerous cell lines was estimated. The hit compound showed cytotoxicity in the MTT test disrupt the normal cytoskeleton of the cell in the pattern similar to that for combretastatin A-4, and preserved its cytotoxic activity after encapsulation in Ca2+ or Mg2+ cross-linked alginate nanocontainers. However, as an article in pharmaceuticals, I have some comments on this manuscript:
1. In the introduction part, the authors emphasized the importance of tubulin and microtubule targeting agents in anti-tumor therapy. Therefore, the compounds synthesized in this work would have tubulin or microtubule targeting properties. The author concluded that compounds disrupt the normal cytoskeleton of the cell in the pattern similar to that for combretastatin A-4 according to fluorescent microscopy data. I think the authors should provide more direct evidence to prove this, not just the comparison of cell morphology in the fluorescent images c with combretastatin A-4. In addition, the fluorescent images are not clear enough, and color images are more appropriate.
2. What is the purpose of drug encapsulation in this work? What’s more, the manuscript lacks a series of characterization of compound encapsulation, including the morphology and size of nanocarriers, the encapsulation rate and loading rate of compound in nanocontainers, and the cytotoxicity and cytoskeletal disruption of the compounds before and after encapsulation.
Author Response
Dear Reviewer,
We thank you for the attention to our work and valuable remarks. We have revised the work, taking into account your comments, and the answers to your remarks are included below. The revised manuscript with our corrections is uploaded.
Yours sincerely,
Dr. Elena B. Averina
Comments and Suggestions for Authors
In this study, a series of Verubulin analogues were obtained and their cytotoxicity towards cancer and non-cancerous cell lines was estimated. The hit compound showed cytotoxicity in the MTT test disrupt the normal cytoskeleton of the cell in the pattern similar to that for combretastatin A-4, and preserved its cytotoxic activity after encapsulation in Ca2+ or Mg2+ cross-linked alginate nanocontainers. However, as an article in pharmaceuticals, I have some comments on this manuscript:
- In the introduction part, the authors emphasized the importance of tubulin and microtubule targeting agents in anti-tumor therapy. Therefore, the compounds synthesized in this work would have tubulin or microtubule targeting properties. The author concluded that compounds disrupt the normal cytoskeleton of the cell in the pattern similar to that for combretastatin A-4 according to fluorescent microscopy data. I think the authors should provide more direct evidence to prove this, not just the comparison of cell morphology in the fluorescent images c with combretastatin A-4. In addition, the fluorescent images are not clear enough, and color images are more appropriate.
Thank you for the note about unclear images. We improved the resolution of images and made them in color.
In our opinion, Figure 2 clearly enough shows the blurring of cytoskeleton, that reflects depolymerization of microtubules under the treatment by derivatives 2b, 2c and 2e, but not by 2a. For compound 2a and control, intact microtubules can be seen ("wires" on image). Сombretastatin A-4 was applied only as non-Verubulin control of cytoskeleton disruption.
In addition to fluorescence microscopy, the independent assay to confirm, that the synthesized compounds target tubulin, was tubulin polymerization assay in vitro – results are described in the manuscript (Figure 3 and Figure S2, and accompanying text).
- What is the purpose of drug encapsulation in this work? What’s more, the manuscript lacks a series of characterization of compound encapsulation, including the morphology and size of nanocarriers, the encapsulation rate and loading rate of compound in nanocontainers, and the cytotoxicity and cytoskeletal disruption of the compounds before and after encapsulation.
Answer:
Encapsulation of the drug into alginate-based nanocontainers was carried out as an attempt to improve a cytotoxicity profile of compound. Besides, it is an approach to obtain a water-soluble form of the drug based on aminopyrimidine and to prevent dispersion of the drug throughout the body and minimize its general toxic effect.
The morphology of nanocontainers was determined in our previous work [https://doi.org/10.1016/j.mencom.2022.09.007]. TEM was used to visualize microgels (nanocontainers). In the field of view of the microscope, gray spherical particles were detected in the case of both nanocontainers. The average diameters of individual particles were found to be 170 nm and 160 nm for (Alg)–Са2+ and (Alg)–Mg2+, respectively.
In this work, the stability of the loaded compound was evaluated under conditions simulating the loading process. It was shown that under these conditions the structure of the incorporated compound is preserved.
A calibration curve (Figure S4) was plotted to quantify the 2c content in the nanocontainers. It was established that both (Alg)–Са2+ and (Alg)–Mg2+ bound 16 wt. % aminopyrimidine 2c which corresponds to the quantitative binding of 2c.
The cytotoxicity of the nanocontainers without 2c is shown in Table 2 for doses of empty nanoparticles of the same volume, which correspond to the studied concentrations of drug 2c in loaded nanocontainers or in free 2c for comparison.
The necessary changes were made in the manuscript and Supplementary material.

Reviewer 2 Report
The synthesis and SAR of Verubulin (Azixa) analogues with increased saturation and further encapsulation in biocompatible nanocontainers is reported
1. As per the reported results, the presented research seems to be promising
2. The synthesis and structural analysis found is limited to NMR studies only what about FTIR and Mass spectroscopy
3. Any characterizations were performed for nanocontainers?
4. Improve the resolution of figures
5. Correct the grammar and sentence error wherever required
Moderate english editing is required
Author Response
Dear Reviewer,
We thank you for the attention to our work and valuable remarks. We have revised the work, taking into account your comments, and the answers to your remarks are included below. The revised manuscript with our corrections is uploaded.
Yours sincerely,
Dr. Elena B. Averina
Comments and Suggestions for Authors
The synthesis and SAR of Verubulin (Azixa) analogues with increased saturation and further encapsulation in biocompatible nanocontainers is reported
- As per the reported results, the presented research seems to be promising
- The synthesis and structural analysis found is limited to NMR studies only what about FTIR and Mass spectroscopy
Answer:
The obtained compounds have quite trivial structure and, in our opinion, NMR and HRMS studies are enough to characterize them. Nevertheless we added description of MS and FTIR of representative compounds 2a-f to the manuscript.
- Any characterizations were performed for nanocontainers?
Answer:
The morphology of nanocontainers was determined in our previous work [https://doi.org/10.1016/j.mencom.2022.09.007]. TEM was used to visualize microgels (nanocontainers). In the field of view of the microscope, gray spherical particles were detected in the case of both nanocontainers. The average diameters of individual particles were found to be 170 nm and 160 nm for (Alg)–Са2+ and (Alg)–Mg2+, respectively.
The amount of loaded substance as well as the encapsulation efficiency was evaluated. The necessary changes were made in the manuscript.
- Improve the resolution of figures
Answer:
Thank you for the note about unclear images. We corrected resolution of image.
- Correct the grammar and sentence error wherever required
Answer:
We have improved the grammar.

Reviewer 3 Report
In this study the authors synthesized a series of analogues of verubulin which is known as a potent tubulin polymerization inhibitor. They explored the cytotoxicity of these compounds against cancer and non-cancer cell lines. With microscopy data, the authors demonstrated these small molecules could disrupt the microtubule assembly in cancer cells. With molecular dynamics simulation, the authors confirmed the docking of selected compound with tubulin heterodimer. Finally, the authors encapsulated the compound in Ca2+ or Mg2+ crosslinked alginate nanoparticles, and tested their cytotoxicity against different cells.
Generally, the work presented is comprehensive in small molecules synthesis and screening, and with broad audience interests, especially those in the field of tumor chemotherapy. However, there are some critical flaws in the data interpretation and experiment design, making current version inappropriate to publish in the journal Pharmaceutics. The reviewer would recommend rejection. Below are a few concerns.
Major concerns:
1) Current data (table 1) does not support the selectivity of these compounds against tumor cells. It is noticeable that for most of compounds tested, the IC50 against non-tumor cells (VA13, HEK293T) is lower or similar to that against tumor cells (MCF7, A549). This indicates a higher or comparable cytotoxicity against non-tumor cells, which defeats the purpose of designing an anti-tumor molecule.
2) Based on the data provided (table 1, figure 3), none of the new synthesized molecules have higher tumor cytotoxicity or stronger inhibition in tubulin polymerization than the current control Verubulin. This greatly weakens the significance of this study.
3) The study for loading compound in alginate nanocarriers is not well designed. Neither nanocontainer is well characterized, nor we know anything about drug loading amount, encapsulation efficiency, and drug release of the compound in alginate nanosystems.
4) It is recommended to provide all the drug dose response curves for the tested compounds in different cells. With current single IC50 value in the table a lot of information is lost.
Minor comments:
1) Introduction needs include general background in using alginate as delivery system, which is a key part in this study.
2) Line 115: It would be better to split 2.2 chapter into multiple parts, as the following discussions include bioactivity, SAR, microtubule disruption, tubulin assembly, and dynamic simulation. With current structure, It is for readers to follow.
3) Line 181: In Fig. 3, why would 2c and 2e lead to higher absorbance at late timepoint? Is 355 nm specific to tubulin synthesis?
4) Line 185: Dose response curves are missing.
5) Line 272: How is the dispersity of alginate nanocarriers? Do they vary differently in different conditions?
6) Table 3: How is IC50 value determined for alginate nanoparticles (entry 2, 4)? Does the uM stand for concentration of alginate polymer or alginate repeating units?
Author Response
Dear Reviewer,
We thank you for the attention to our work and valuable remarks. We have revised the work, taking into account your comments, and the answers to your remarks are included below. The revised manuscript with our corrections is uploaded.
Yours sincerely,
Dr. Elena B. Averina
Comments and Suggestions for Authors
In this study the authors synthesized a series of analogues of verubulin which is known as a potent tubulin polymerization inhibitor. They explored the cytotoxicity of these compounds against cancer and non-cancer cell lines. With microscopy data, the authors demonstrated these small molecules could disrupt the microtubule assembly in cancer cells. With molecular dynamics simulation, the authors confirmed the docking of selected compound with tubulin heterodimer. Finally, the authors encapsulated the compound in Ca2+ or Mg2+ crosslinked alginate nanoparticles, and tested their cytotoxicity against different cells.
Generally, the work presented is comprehensive in small molecules synthesis and screening, and with broad audience interests, especially those in the field of tumor chemotherapy. However, there are some critical flaws in the data interpretation and experiment design, making current version inappropriate to publish in the journal Pharmaceutics. The reviewer would recommend rejection. Below are a few concerns.
Major concerns:
- Current data (table 1) does not support the selectivity of these compounds against tumor cells. It is noticeable that for most of compounds tested, the IC50 against non-tumor cells (VA13, HEK293T) is lower or similar to that against tumor cells (MCF7, A549). This indicates a higher or comparable cytotoxicity against non-tumor cells, which defeats the purpose of designing an anti-tumor molecule.
- Based on the data provided (table 1, figure 3), none of the new synthesized molecules have higher tumor cytotoxicity or stronger inhibition in tubulin polymerization than the current control Verubulin. This greatly weakens the significance of this study.
Answer for 1,2:
Lack of selectivity definitely is a common feature of reported Verubulin analogues as well as other tubulin-targeting agents. We made an attempt to overcome it by increasing the molecule saturation and non-planarity. Though this attempt was not successful, in our opinion, the obtained results are worth reporting, as they describe novel direction of Verubulin structure modification and make an important contribution to SAR of Verubulin analogues.
Besides, though in the terms of IC50, we cannot observe selectivity of the studied compounds towards cancer cells, drug dose response curves reveal the difference in viability of VA13 and faster growth cell lines MCF7' and A549 as exemplified with Verubulin (1) itself and compounds 2c, 2e, 2m. This pattern of cytotoxicity is common for tubulin-disrupting drugs in our experience. The full graphs of dependencies concentration-viability were added to manuscript.
3) The study for loading compound in alginate nanocarriers is not well designed. Neither nanocontainer is well characterized, nor we know anything about drug loading amount, encapsulation efficiency, and drug release of the compound in alginate nanosystems.
Answer:
The morphology of nanocontainers was determined in our previous work [https://doi.org/10.1016/j.mencom.2022.09.007]. TEM was used to visualize microgels (nanocontainers). In the field of view of the microscope, gray spherical particles were detected in the case of both nanocontainers. The average diameters of individual particles were found to be 170 nm and 160 nm for (Alg)–Са2+ and (Alg)–Mg2+, respectively.
The amount of loaded substance as well as the encapsulation efficiency was evaluated.
The necessary changes were made in the manuscript.
4) It is recommended to provide all the drug dose response curves for the tested compounds in different cells. With current single IC50 value in the table a lot of information is lost.
Answer:
The full graphs of dependencies concentration-viability were added to Supplementary Materials (Figure S1)
Minor comments:
- Introduction needs include general background in using alginate as delivery system, which is a key part in this study.
Answer:
Information about alginate as a delivery system was included in the Introduction. Necessary changes were made in the manuscript.
- Line 115: It would be better to split 2.2 chapter into multiple parts, as the following discussions include bioactivity, SAR, microtubule disruption, tubulin assembly, and dynamic simulation. With current structure, It is for readers to follow.
Answer:
We have split the chapter as required.
3) Line 181: In Fig. 3, why would 2c and 2e lead to higher absorbance at late timepoint? Is 355 nm specific to tubulin synthesis?
Answer:
The absence of a decrease in light absorption in samples with substances, as well as in samples with verubulin, indicates a disruption in the dynamics of microtubules - the establishment of equilibrium between the processes of polymerization-depolymerization (as occurs in the control).
Absorption at λ=355 nm is specific for assay of tubulin polymerization (not synthesis). Below we have provided a number of links confirming this.
Lee, James C.; Timasheff, Serge N. (1975). Reconstitution of microtubules from purified calf brain tubulin. Biochemistry, 14(23), 5183–5187. doi:10.1021/bi00694a025
Lee, James C.; Timasheff, Serge N. (1977). In vitro reconstitution of calf brain microtubules: effects of solution variables. Biochemistry, 16(8), 1754–1764. doi:10.1021/bi00627a037
Shelanski ML, Gaskin F, Cantor CR. Microtubule assembly in the absence of added nucleotides. Proc Natl Acad Sci U S A. 1973 Mar;70(3):765-8. doi: 10.1073/pnas.70.3.765.
https://www.cytoskeleton.com/hts-bulk/tubulin-and-maps/bk006p
4) Line 185: Dose response curves are missing.
Answer:
We have inserted the curves in Supplementary Materials (Figure S2)
5) Line 272: How is the dispersity of alginate nanocarriers? Do they vary differently in different conditions?
Answer:
The polydispersity of alginate-based nanocarriers was evaluated by means of DLS using PDI criterium. Necessary changes were made in the manuscript.
6) Table 3: How is IC50 value determined for alginate nanoparticles (entry 2, 4)? Does the uM stand for concentration of alginate polymer or alginate repeating units?
Answer:
For alginate nanocontainers (Alg)–Ca2+ and (Alg)–Mg2+, we described in more detail the conditions for obtaining IC50 values and survival curves of the studied cells. Survival curves for free 2c and 2c in nanocontainers (2c∙HCl–(Alg)–Ca2+ and 2c∙HCl–(Alg)–Mg2+) were obtained for concentrations of the active substance 2c, the final concentrations of which when added to cells were 0.8-25 µM. Since all the used nanocontainers (2c∙HCl–(Alg)–Ca2+, 2c∙HCl–(Alg)–Mg2+, as well as empty samples (Alg)–Ca2+ and (Alg)–Mg2+ )were identical in the composition of COO- - groups in the stocks (CCOO- 1000μM), then the final concentrations of COO- - groups for the nanocontainers with 2c introduced into the experiment (2c∙HCl–(Alg)–Ca2+ and 2c∙HCl–(Alg)–Mg2+) were 8-250 µM. The same doses with the same concentrations of COO- - groups for empty nanoparticles were used in the experiment - the final concentrations of COO- - groups for the nanocontainers (Alg)–Ca2+ and (Alg)–Mg2+) introduced into the experiment were 8-250 μM. Thus, we corrected the dimension of the concentration axis and the dimension of the IC50 determination for empty nanocontainers administered in doses similar to those of 2c-loaded samples. This is shown in the Manuscript and in the Supplementary.

Round 2
Reviewer 1 Report
Accept.
Reviewer 3 Report
The authors address the questions well and i recommend it ready for publication.